# A Preliminary Investigation: Evaluating the Effectiveness of an Occupational Specific Training Program to Improve Lower Body Strength and Speed for Law Enforcement Officers

**DOI:** 10.3390/ijerph18147685

**Published:** 2021-07-20

**Authors:** Ian Bonder, Andrew Shim, Robert G. Lockie, Tara Ruppert

**Affiliations:** 1Department of Kinesiology & Exercise Science, College of Saint Mary, Omaha, NE 68106, USA; ashim@csm.edu; 2Department of Kinesiology, California State University, Fullerton, Fullerton, CA 92831, USA; rlockie@fullerton.edu; 3Department of Occupational Therapy, College of Saint Mary, Omaha, NE 68106, USA; truppert@csm.edu

**Keywords:** strength, power, hex bar deadlift, sprint, speed, law enforcement

## Abstract

Based on current law enforcement officer (LEO) duties, musculoskeletal injury risk is elevated due to the unpredictable nature of physically demanding tasks. The purpose of this 4-week study was to determine the effectiveness of a 15-min post-shift standardized occupational specific training program. The standardized program was designed to improve lower-body strength and speed to aid physically demanding task performance. Seven male LEOs completed the program after their 12-h shift. Subjects were required to use the department fitness center to perform the 15-min standardized program consisting of a dynamic warm-up, 4 sets of 3 repetitions on hex-bar deadlift and four 20-m sprints. Two minutes of rest was required between each set of 3 repetitions on hex-bar deadlift and 1 min of rest between each 20-m sprint. A dependent T-test was used between pre-test and post-test scores for hex-bar deadlift (HBD) and sprint. Data revealed significant improvements in relative lower-body strength with HBD (*p* ≤ 0.001). However, insignificant results were demonstrated with the 20-m sprint (*p* ≤ 0.262). In conclusion, a 15-min post-shift workout can improve lower-body strength as measured by the hex-bar deadlift. However, data indicated running speed may require a different training approach to improve the 20-m sprint.

## 1. Introduction

Present day law enforcement officers (LEOs) face an increased risk of injury due to job-related tasks of a varied, unpredictable, and highly physically demanding nature [1]. Examples of job-related tasks include pushing, pulling, lifting, dragging, foot pursuits and use of physical force [2,3]. In addition to the physical demands of police work, LEOs are also exposed to the possibility of long-term chronic injury due to the amount of time spent sitting in their police cruiser. A previous study by Gruevski et al. [4] reported that nearly 6 h of a LEO 12-h shift is spent sitting in their cruiser and 40% of those 6 h in the cruiser is spent using their Mobile Data Terminal. This can cause the officers to twist and contort their bodies in uncomfortable positions for long periods of time.

Due to a combination of both the physical and sedentary nature of law enforcement duties [5,6,7], active duty LEOs are faced with the possibility of experiencing a wide array of musculoskeletal injuries. The possibility of sudden physical action coupled with sedentary behavior, i.e., time spent sitting in a police cruiser, puts LEOs at an increased risk of injury [8]. Musculoskeletal injuries occur through excessive stress and could lead to overuse injuries affecting tendons, ligaments, muscles and bone [9]. In turn, 40% of those musculoskeletal injuries are the result of exertional lower body musculoskeletal injuries, accounting for the observation of approximately 75% of limited duty days in LEOs [10]. Injury rates vary across different law enforcement agencies (LEAs) with a reported range of 240 to 2500 injuries per 1000 law enforcement personnel per year, with the most commonly reported injury sites including the broad categories of “upper extremity injuries” (i.e., the torso and the back) [1]. A separate study focused on 353 Korean police officers reported the highest incidence of musculoskeletal injury in the shoulder (44.2%), waist (41.4%) and neck (31.2%) [11]. Costs related to injuries in the law enforcement community have been found to range from $2500 up to $12,000 (USD) [1]. In addition to the financial burden related to injuries in law enforcement, the U.S. Department of Labor and Statistics cites that protective services employees miss 15 days of work per year on average due to injury [12].

An increased length of service to the police department has been associated with recurrent and chronic episodes of low back pain in LEOs [13]. A large cross-sectional study conducted throughout the Quebec Police Department found that nearly 68% of respondents to a web-based survey reported incidents of low back pain in the past 12 months, and of that 68%, nearly 97% had the perception that their low back pain was related to their work on the police force [14]. Causes for low back pain range from little lumbar support and wearing a duty belt [15] to time spent driving, both the number of hours driven and the distance traveled [16]. Nearly 25% of LEOs are considered prolonged drivers, driving more than 25,000 km/year, with an estimated 18% of those drivers experiencing low back pain while driving [15]. Sedentary time, particularly those hours spent sitting in the cruiser along with working on the Mobile Data Terminal, puts LEOs at an increased risk for developing chronic low back pain [17] due to an increased amount of prolonged lumbar flexion [4] leading to decreases in levels of physical activity.

A decreased level of physical activity could have negative impacts on the overall strength and lower-body power of LEOs, which in turn could affect the officers’ ability to successfully pursue a suspect on foot, aid in obstacle clearance and improve victim/body drag capabilities [18,19]. Strength is a concept of maximal force output in opposition to power, which is a measurement of velocity and how quickly force can be applied to move an external object [20]. Both strength and power are underlying components of everyday requirements of LEOs including foot pursuits, obstacle clearance, dragging, jumping, pushing and pulling. Measures of both absolute and relative lower-body strength, as it relates to the deadlift, have been shown to have statistically significant correlations with a dummy drag and pack march while other studies have shown lower-body strength and power are essential for various occupational performance attributes such as sprinting varied distances, climbing stairs and jumping [21,22]. Furthermore, Orr et al. [21] suggested that both absolute and relative strength may improve task performance in LEOs.

In regards to power development and its relationship to sprint speed improvements, several studies observing measures of lower-body power through the implementation of various jump tests, have been shown to correlate directly with sprint speed over various distances in both athletic populations and SWAT officers [2]. Two previous studies reviewed the effects of a hexagonal bar deadlift (HBD) and its relationship to peak power in a body drag [21,23]. Several previous studies observing LEOs used various types of vertical jump tests to determine lower body power [2,22,23,24,25,26]. Furthermore, two separate studies investigated the relationship between 10 and 20-m sprint speed and their relationships to load-carriage tasks and the 75-yard pursuit run, respectively [27,28]. However, no studies were found to investigate the concurrent use of an HBD 3-repetition maximum (3 RM) and 20-m sprints to improve overall lower body strength and speed. Improvements in lower-body strength and sprint speed may aid in improved foot pursuit ability, obstacle clearance, victim/body drag capabilities, and suspect apprehension [27]. Given the time demands faced by LEOs, a concurrent training program to improve lower-body strength and speed may prove valuable for the law enforcement community. Therefore, the purpose of this study is to evaluate the effectiveness of a standardized 15-min post-shift occupationally specific training program to improve both lower-body strength and speed. The investigators hypothesize that the 15-min post-shift training intervention will produce significant improvements in HBD 3 RM and overall 20-m sprint times within four weeks.

## 2. Materials and Methods

### 2.1. Subjects

A sample size of convenience of seven male LEOs from a local LEA were recruited to participate in this study. All participants were 26–47 years old (age 31 ± 7.33 years; weight 96.03 ± 9.47 kg; height 181.79 ± 5.65 cm; BMI 29.1 ± 2.6 kg/m^2^). Subject participation in the study was voluntary and recruitment of the participants took place during the early morning line-up by the line-up Sergeant. Recruitment was conducted by handing out a word-of-mouth informational flier detailing the study participant recruitment information. All participants filled out a PAR-Q health history questionnaire prior to beginning the study to determine if they were eligible for inclusion. All participants who were free of any musculoskeletal disorders and injuries prior to taking part in the study were deemed as healthy and physically fit to participate. All participants completed an informed consent, which detailed the purpose and objective of the study as well as their rights to withdraw at any time, prior to taking part in this research. The Institutional Review Board at a Midwest University approved the study protocol (CSM 2008).

### 2.2. Procedures

The data in this study were collected by the primary and secondary investigator. Participant data was kept and tracked in a Microsoft Excel spreadsheet under a unique identifier number known only to the participant, primary investigator and secondary investigator. Instruction on how to complete the brief, standardized dynamic warm-up before starting the initial performance assessment on both the HBD and 20-m sprint were conducted on all participants. Initial performance assessment data were tracked and recorded for use as comparison with final performance assessment data. All LEOs taking part in this study were trained by a Certified Strength and Conditioning Specialist. An instructional video on how to perform the standardized dynamic warm-up, HBD, and 20-m sprint were recorded prior to beginning the study for all LEOs to use as a reference to check both correct form and technique on the HBD and 20-m sprint. All participants wore athletic shoes for initial and final performance assessments.

### 2.3. Training Intervention

Upon completion of the initial performance assessment and data collection, LEOs followed a pre- or post-shift 3 day per week protocol including the brief, standardized dynamic warm-up, 4 sets of 3 repetitions on the HBD and four 20-m sprints for the entirety of the 4-week study. Each of the sessions were to take no longer than 15 min. All study participants were instructed to check-in and check off the 3 days they completed their pre- or post-shift protocol on a check-in sheet located in the LEA workout facility. This sheet was collected at the beginning of the following week by the primary investigator to ensure participant compliance during each week of the study. During each of the 12 training sessions, 4 total sets of 3 repetitions with 2 min of rest between each set was completed for the HBD. Set 1 of the HBD was completed using 30% of baseline 3RM testing, set 2 at 50% of 3 RM, set 3 at 70% of 3RM and set 4 at 90% of 3 RM. During each of the 12 training sessions, upon completion of 4 sets of 3 repetitions on the HBD, four 20-m sprints were to be completed. One minute of rest was to be allotted between each of the 20-m sprints. After completion of 12 total sessions, including the initial performance assessment, the LEOs that successfully completed all sessions were taken through a final performance analysis assessment. Prior to data collection during the final performance assessment, participants were to complete their brief, standardized dynamic warm-up. During the final performance assessment, participants repeated the exact same initial performance assessment protocol of 4 sets of 3 repetitions on the HBD with 2 min of rest between attempts on a rubber training surface to determine a final HBD 3RM. Upon completion of testing on the HBD, participants completed 4 timed 20-m sprints indoors at the same location, indoors on carpet, to determine a final, personal best 20-m sprint time.

### 2.4. Hex-Bar Deadlift

Relative lower-body strength was assessed via an HBD 3RM where participants performed a HBD (standard 45-pound bar) for 3 repetitions using the most weight possible with correct form and technique. The HBD 3RM was assessed in both the initial and final performance assessments. HBDs were performed with the participant standing inside the hex-bar with both feet aligned hip to shoulder width apart and toes slightly turned out. To begin the lift, participants were to drop their hips, keep the chest up, back flat and grab each of the handles. Once achieving a starting position, holding on to the hex-bar, participants were instructed to perform the repetition, standing up while pulling the hex bar up from the ground. When pulling the hex bar off the ground, participants were to keep their chest up and have their hips, chest and shoulders going up at the same rate. Participants then returned the bar to the ground, keeping the chest up, under control, and refraining from “bouncing” the plates off of the ground and in to the next repetition. Three successive repetitions were performed in this manner. If the weight was too heavy and form was broken, the participants were instructed to return the bar to the ground and to decrease the total weight on the bar by 5–10 pounds. When participants completed 3 successful repetitions with correct form, 5–10 pounds was added to the hex-bar. Two minutes of rest was allotted between each of the 4 sets in the initial and final performance assessments.

### 2.5. 20-m Sprint

Sprint speed was assessed via a 20-m sprint. 20-m sprint times were assessed in both the initial and final performance assessments with the use of a DashR Kit (DashR Systems, Lincoln, NE, USA) laser timing device connected by Bluetooth technology to the DashR App. Twenty meters was marked off inside the LEA with both a start and end line. Each 20-m sprint performed was to start in a 3-point stance position. The 3-point stance was to have both feet staggered, one foot forward and one foot back, opposite hand of the forward foot up on the finger tips, front knee driven forward over the front foot, front foot flat, and back foot up on the toe with the heel off the ground. A 3-point stance was used due to the starting biomechanical pattern allowing for increased angular momentum when compared with a standing stance [29]. The angle of the torso and chest were to be in alignment with the angle of the front shin. Participants were instructed to run as fast as they could from the start line through the end line. One minute of rest was allowed between each 20-m sprint in the initial and final performance assessments.

### 2.6. Statistical Analysis

Statistical analysis was performed using a dependent two-tailed T test (SPSS Version 26) (IBM, Armonk, NY, USA) to compare differences between the initial and final performance assessments. Both means and standard deviations were calculated for results in the HBD 3RM and 20-m sprint for the initial and final performance assessment.

## 3. Results

Comparative data for both the initial and final performance assessment are listed in Table 1. The data for HBD 3RM indicated significant improvements in lower-body strength (*p* ≤ 0.001), with the 7 participants experiencing an average increase of 4.5% weight lifted. Figure 1 provides a graphic representation of HBD 3RM initial performance assessment and final performance assessment results for each of the 7 participants. However, significant improvements were not noted (*p* ≤ 0.262) in 20-m sprint times. Four of the seven participants improved their overall 20-m sprint times while the other three participants experienced slight increases in 20-m sprint times. The largest decrease in 20-m sprint time was 0.17 s while the largest increase in 20-m sprint time was 0.06 s (see Figure 2). As a group, the seven participants experienced an average decrease of 1.2% in sprint time. All participants improved their HBD 3RM over the course of this 4-week study. Improvements ranged in increases of 10–20 pounds between the initial performance assessment and final performance assessment.

## 4. Discussion

This study provided an analysis of the effectiveness of an occupationally specific training program to improve lower body strength and speed for LEOs. Both HBD 3RM and 20-m sprints were assessed separately. Each component was compared once completed to determine if any significant results were noted. All participants in this study were active duty, male LEOs. The results indicate that maximal relative lower-body strength as measured by the HBD 3RM could be significantly improved by the occupational specific training program; however, this was not the case for the 20-m sprint. With regards to physically demanding job-related tasks, this occupationally specific training program may provide a basis to influence physical training to improve lower-body strength for LEOs, specifically in situations where time is a limiting factor.

HBD exercises have been identified as a congruent method for improving critical job task performance common to LEO daily duties [23,30]. Lockie et al. [23] demonstrated how the implementation of HBD exercises strongly correlated with physically demanding activities for LEOs. Based on these findings, the investigators felt the need to implement HBD exercises in a convenient time-frame and manner. Furthermore, lower-body strength, particularly relative strength, has been observed to contribute to the sustained maintenance of proficiency in physically demanding, job related tasks [27]. A focus on total body absolute and relative strength development has also been suggested as a vital component in a LEO’s strengthening and reconditioning programs for injured officers [21]. Training to improve relative lower-body strength would have implications of increasing overall volume load and repetitions when compared with training to improve absolute strength. However, the lower intensity of weight used could possibly have a protective and regenerative effect on low back health, thus potentially improving LEO longevity and sustained performance in physically demanding tasks. By observing Figure 1, each subject significantly increased their lower-body relative strength from the pre-test to the post-test (*p* ≤ 0.001). No injuries or discomfort were reported during the 4-week training intervention period, thus demonstrating the value of these results from the training intervention.

However, when the goal is to produce improvements in speed, such as improving 20-m sprint times, which would aid in the task of foot pursuits, other training means must be considered. Of the seven participants taking part in this study, four saw improved sprint times with the largest improvement being a decrease of 0.17 s. The remaining three participants experienced slight increases in 20-m sprint times with the largest increase being 0.06 s. Sprinting is a complex task potentially requiring different training modalities and regimes in addition to more time needed to observe improvements. A study by Delecluse [31] noted that sprint performance is composed of several phases (initial acceleration, transition phase, maximum running speed), with each phase requiring a specific training modality to elicit further improvement. The utilization of separately focused training blocks dedicated to improving each phase of sprint performance, could allow for further improvements to be noted when the goal is to improve individual speed. A longer training time-frame may also allow for increased sprinting ability. A 6-week study by Gil et al. [32] noted a 3% overall improvement for 20-m sprint times. Furthermore, a study by Pareja-Blanco et al. [33] utilized an 8-week concurrent training study to elicit significant improvements in sprinting over varied distances. Thus, extending the length of training to 6 weeks or longer may promote further improvements in sprinting ability.

The differences in biomechanical patterns between HBD and sprinting are great enough that the tasks did not produce force improvements in the same direction, which in-turn led to significant improvements in only one of the two variables assessed. This is in-line with Franklin Henry’s Theory of Specificity [34]. Henry’s Theory of Specificity postulates that specific improvements in motor skills are the result of performing specific tasks [35]. Thus, skills learned from one specific task and then transferred to a similar task are more easily, and possibly more quickly, reproduced in future movements leading to higher levels of proficiency. Use of the HBD in training has been shown to reduce horizontal bar displacement away from the body, promoting greater average vertical force development [36]. In contrast, improvements in horizontal force are necessary to improve sprinting ability [37]. Improvements in 20-m sprint may have been minimized due to a lack of strength training to improve horizontal force production within this study. Examples of movements to improve horizontal force production and align more specifically with speed development include conventional barbell deadlifts to increase horizontal bar displacement in relation to the body [36] and back squatting at light to moderate loads in combination with light-load sled towing using 12.5% body weight during a resisted sprint [33]. Additional factors that could have led to minimal improvements in 20-m sprint times include sprint testing and training being conducted on carpet and external cues before beginning sprint testing not being as detailed as were those for the HBD 3RM.

## 5. Conclusions

This study has some limitations that merit acknowledgement. The sample size was small (N = 7), however, it did provide for an initial assessment of the effects of a short-term, short-duration strength and sprint training program. Nonetheless, larger sample sizes should be utilized in future studies analyzing the impacts of lower-body maximal relative strength training on speed development. Secondly, this study did not have a control group, which could have proved beneficial in comparison of results between the initial performance assessment and final performance assessment. Additionally, a 4-week length of time for the study may not have been long enough to promote significant improvements in sprint speed. As mentioned previously, significant improvements in sprint performance have been observed in studies 6–8 weeks in length. Furthermore, research should allow for training on more days of the week to follow a daily undulating periodization approach, as this method has been shown to be superior for enhancing maximal strength development and requires the trainee to continuously have to adapt to changing stressors [38]. The daily undulating periodization approach would allow for maximal relative strength training and development on one day and power and speed development to be trained on a separate day. Future research should be directed at how to incorporate both maximal absolute and relative strength training and lower-body power and speed development into a time-efficient, occupationally specific training program to aid in improved foot pursuit ability for LEOs.

In conclusion, this preliminary analysis showed that maximal relative lower-body strength, as measured by the HBD 3RM, could be significantly improved in a time-efficient manner by utilizing the occupationally specific training program outlined in this study. Even though improvements were not shown for all subjects in the 20-m sprint, future research on instruction and cueing in the 20-m sprint while training over a longer period of time could produce different outcomes. This training intervention demonstrates practicality with limited space, equipment, and time investment. However, applications of these exercises could be implemented in a clinical setting for future studies with tactical occupations.

## Figures and Tables

**Figure 1 ijerph-18-07685-f001:**
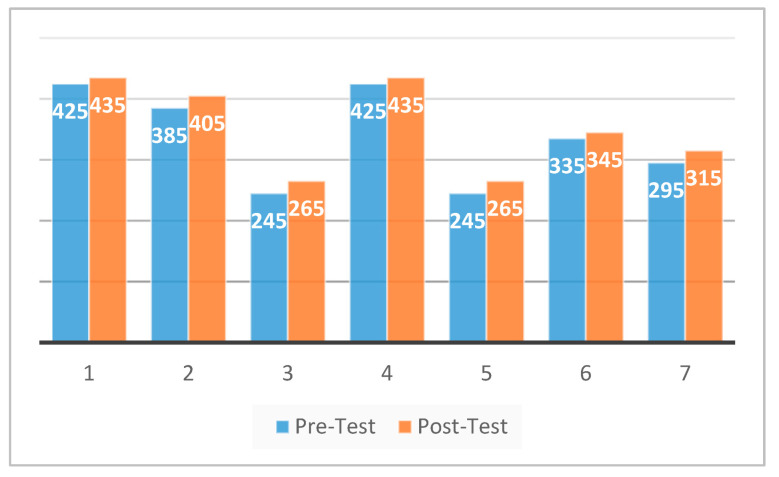
HBD 3RM 4-week testing results. (*p* ≤ 0.001).

**Figure 2 ijerph-18-07685-f002:**
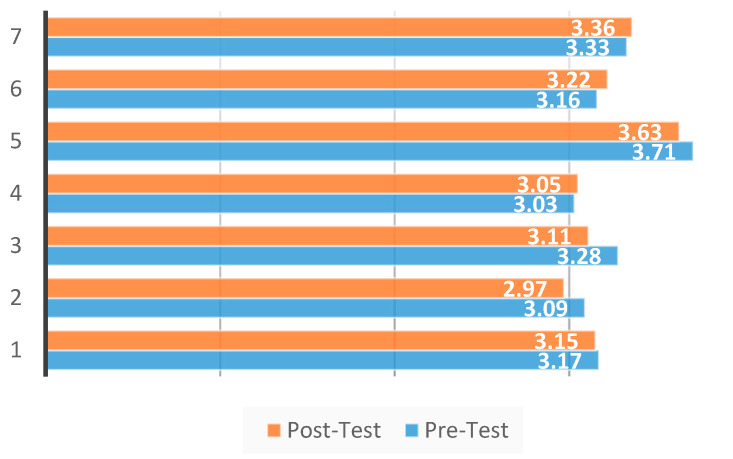
4-week 20-m sprint times results (*p* ≥ 0.05).

**Table 1 ijerph-18-07685-t001:** Initial and final performance assessment HBD 3RM and 20-m sprint times.

Assessment	HBD 3RM (Pounds)	20-m Sprint (Seconds)
Initial	336.43 ± 77.98	3.25 ± 0.23
Final	352.14 ± 74.32	3.21 ± 0.22

## Data Availability

The data presented in this study are available on request from the corresponding author. The data are not publicly available to keep the identity of those participating law enforcement officers involved in the study private.

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
