# Peer review of "A Preliminary Investigation: Evaluating the Effectiveness of an Occupational Specific Training Program to Improve Lower Body Strength and Speed for Law Enforcement Officers"

_ijerph, 2021, doi:10.3390/ijerph18147685_

Round 1

Reviewer 1 Report

Interesting and worthwhile study.
I think they should rethink the title because it's a preliminary study, so they should put this information out.
This is because the sample size was small, only 7 individuals and the duration of the program is very short (4-week). Also the fact that did not have a control group reinforces this idea.
In Statistical Analysis, I think you should remove the last sentence 'P-value for HBD 3RM was < .001 while p-value for the 20-meter sprint was < .262, because it is, and well framed, in the next field.

I think that effectively the study is correct in terms of state of the art and methodology, however, it has a very small sample and a very short program implementation time. I think that as a preliminary study it is interesting to publish. However, the authors should, in my opinion, extend the sample and duration of the program. The literature review is current and directed, the methodological issues are well formulated. The results are well presented and discussed. I have no issues for improvement to point out beyond the few I have suggested. I leave the publication and the need to adapt the title to the editor's consideration.

Author Response

Interesting and worthwhile study.
I think they should rethink the title because it's a preliminary study, so they should put this information out. This is because the sample size was small, only 7 individuals and the duration of the program is very short (4-week). Also the fact that did not have a control group reinforces this idea. Thank you. We have changed the title of the study according to your recommendation to be "A Preliminary Investigation: Evaluating the Effectiveness of an Occupational Specific Training Program to Improve Lower Body Strength and Speed for Law Enforcement Officers"

In Statistical Analysis, I think you should remove the last sentence 'P-value for HBD 3RM was < .001 while p-value for the 20-meter sprint was < .262, because it is, and well framed, in the next field. Your suggestion to remove the last sentence has been completed in the manuscript. 

I think that effectively the study is correct in terms of state of the art and methodology, however, it has a very small sample and a very short program implementation time. I think that as a preliminary study it is interesting to publish. However, the authors should, in my opinion, extend the sample and duration of the program. The literature review is current and directed, the methodological issues are well formulated. The results are well presented and discussed. I have no issues for improvement to point out beyond the few I have suggested. I leave the publication and the need to adapt the title to the editor's consideration. Thank you for your time and suggestions in assisting to improve the overall manuscript. 

Reviewer 2 Report

Bonder and Colleagues, provided an analysis of the effectiveness of an occupationally specific training program to improve lower body strength and speed for LEO’s. The manuscript was well written, but I have major concerns to address. 

Introduction

Although the purpose of the current study was correctly addressed, the study's hypothesis is missing. 

Materials and Methods 

Was sample size calculated? Please, explain if the sample size was enough for the conduction of the current study.

Why did the authors not perform a controlled group of the exercise intervention?

Results 

Since the data dispersion and inter-individual response to exercise training is an important matter, I suggest a graph of the inter-individual adaptations on lower body strength and speed following exercise training. 

Discussion 

The authors should address recent references in the discussion section highlighting the importance of this mode of training compared to the others to improve lower body strength and speed.

Conclusion

More practical and clinical applications from the current results should be addressed. 

Author Response

Introduction

Although the purpose of the current study was correctly addressed, the study's hypothesis is missing. A research hypothesis has been added to the end of the introduction.

Materials and Methods 

Was sample size calculated? Please, explain if the sample size was enough for the conduction of the current study. The study authors decided due to the small sample size that a power analysis would not be determined and agreed to revise the title of the manuscript to include, "A Preliminary Study:". This is based on a strong recommendation from Reviewer #1.

Why did the authors not perform a controlled group of the exercise intervention? It was a sample of convenience, the department was a relatively small police force with limited numbers to recruit participants. 

Results 

Since the data dispersion and inter-individual response to exercise training is an important matter, I suggest a graph of the inter-individual adaptations on lower body strength and speed following exercise training. A graph has been added detailing inter-individual responses to 20-meter sprint over 4-week duration. Titled Figure 2. 

Discussion 

The authors should address recent references in the discussion section highlighting the importance of this mode of training compared to the others to improve lower body strength and speed. Current citations were added in the discussion to justify the use of hex bar deadlift. Justification of 20-meter sprint was already acknowledged in the manuscript.  

Conclusion

More practical and clinical applications from the current results should be addressed. 1-2 sentences have been added to the end of the conclusion to address practicality and clinical applications for future research. 

The authors thank you for your time spent reviewing our manuscript and providing positive and constructive feedback.

Round 2

Reviewer 2 Report

Thanks! All my concerns were addressed.